# Zebrafish: A Useful Animal Model for the Characterization of Drug-Loaded Polymeric NPs

**DOI:** 10.3390/biomedicines10092252

**Published:** 2022-09-11

**Authors:** Sara Bozzer, Luca De Maso, Maria Cristina Grimaldi, Sara Capolla, Michele Dal Bo, Giuseppe Toffoli, Paolo Macor

**Affiliations:** 1Department of Life Sciences, University of Trieste, 34127 Trieste, Italy; 2Experimental and Clinical Pharmacology Unit, Centro di Riferimento Oncologico di Aviano (CRO), IRCCS, 33081 Aviano, Italy

**Keywords:** zebrafish, polymeric NPs, doxorubicin

## Abstract

The use of zebrafish (ZF) embryos as an in vivo model is increasingly attractive thanks to different features that include easy handling, transparency, and the absence of adaptive immunity until 4–6 weeks. These factors allow the development of xenografts that can be easily analyzed through fluorescence techniques. In this work, ZF were exploited to characterize the efficiency of drug-loaded polymeric NPs as a therapeutical approach for B-cell malignancies. Fluorescent probes, fluorescent transgenic lines of ZF, or their combination allowed to deeply examine biodistribution, elimination, and therapeutic efficacy. In particular, the fluorescent signal of nanoparticles (NPs) was exploited to investigate the in vivo distribution, while the colocalization between the fluorescence in macrophages and NPs allows following the elimination pathway of these polymeric NPs. Xenotransplanted human B-cells (Nalm-6) developed a reproducible model useful for demonstrating drug delivery by polymeric NPs loaded with doxorubicin and, as a consequence, the arrest of tumor growth and the reduction in tumor burden. ZF proved to be a versatile model, able to rapidly provide answers in the development of animal models and in the characterization of the activity and the efficacy of drug delivery systems.

## 1. Introduction

Although most human pathologies have been modeled using mammalian systems, such as mice, in recent years, attention has focused on the tropical freshwater fish Danio rerio (zebrafish, ZF) as an outstanding tool for studying human diseases [1]. ZF models are now a well-known option for implementing personalized medicine strategies, along with other models of patient-derived xenografts or patient-derived organoids [2,3,4,5,6]. ZF models are small and robust, cheap to maintain, and a single matching produces hundreds of eggs that develop extremely rapidly. An incomparable and unique feature is their optical transparency, which is important for easy visualization of the xenotransplanted cells or the biodistribution of the subject matter of research, or both simultaneously, with the aid of a fluorescence microscope and with high throughput results [7]. 

Various injection sites were tested [8,9,10], but the yolk sac [8,11,12] has been shown to be an ideal approach for localized xenotransplantation in 2-day-old embryos, and in parallel, the Cuvier’s duct is the best option for delivering substances into the embryo’s bloodstream [13]. On this basis, ZF represents an innovative tool in the research landscape to study cancer diseases [14], including pediatric cancers such as pediatric B-cell malignancies. Although the treatment of pediatric B-cell malignancies can be considered a success story, with current overall survival (OS) of ~80% in the United States, the therapy-related side effects are still alarming [15]. For this reason, in the last years, researchers’ interest has more frequently been focused on the development of a strategy that combines the knowledge about drugs with newborn nano-carriers for effective and selective drug delivery. In this context, polymeric NPs with well-defined size and shape, such as those synthesized using the polymers polylactide-*co*-glycolide acid (PLGA) and poly (vinyl alcohol) (PVA) [16] can improve the drug delivery process, thanks to the encapsulation of the drug that protects it until the nano-vector reaches the target through the enhancer permeability and retention (EPR) effect and releases its contents, leading to a reduction in the severe side effects associated with the use of chemotherapeutic agents [17,18]. In this context, ZF is an ideal candidate to rapidly evaluate xenograft tumor development, including the development of a B-cell malignancy model implanted in a large number of animals, and to investigate and potentially compare novel therapeutic approaches during their initial characterization [19].

In the present study, we propose a fluorescent-based quantification method for the setup of a B-cell malignancy model in ZF embryos using Nalm-6 cells, a B-acute lymphoblastic leukemia (ALL)-like cell line. The developed Nalm-6 cell line model in ZF was employed to investigate the capability of PLGA-PVA polymeric NPs to reach the tumor site, as well as the killing capability of doxorubicin-loaded PLGA-PVA polymeric NPs. We found that PLGA-PVA polymeric NPs distribute in the ZF bloodstream and reach the tumor, and doxorubicin-loaded PLGA-PVA polymeric NPs are capable of killing Nalm-6 cells, thus reducing tumor cell burden.

## 2. Materials and Methods

### 2.1. PLGA-PVA Polymeric NPs Synthesis

PLGA-PVA polymeric NPs were produced in our laboratory with small modifications to the Vasir and Labhasetwar protocol [16]. Firstly, the PLGA (Sigma, Saint Louis, MO, USA) solution was prepared by dissolving 30 mg of PLGA in 1 mL of chloroform (Sigma) in a 5 mL glass vial with magnetic stirring. During this process, the solution for the aqueous core of NPs was prepared, and fluorescein-5-isothiocyanate (FITC)-conjugated Bovine Serum Albumin (BSA, Sigma) or doxorubicin (Pfizer Inc. New York, NY, USA) was mixed in Tris-EDTA buffer. Finally, the PVA (Sigma) solution was set; 0.2 g of PVA was sprinkled slowly over 10 mL of cold Tris-EDTA buffer, centrifuged at 200× *g* for 10 min at 4 °C, and 10 μL of chloroform was then added to saturate the PVA solution.

The solution for the core was added to the PLGA solution in two aliquots of 100 μL each, vortexed for 1 min after each addition, and sonicated. This primary emulsion was added in two portions to 6 mL of PVA solution and vortexed for 1 min after each addition. The resulting emulsion was stirred overnight (at RT) to allow chloroform to evaporate. NPs were recovered by ultracentrifugation at 11,000× *g* for 20 min at 4 °C, and the pellet was resuspended in 5 mL of Tris-EDTA buffer. The sample was washed again and resuspended in H_2_O MilliQ filtered 0.2 μm.

### 2.2. PLGA-PVA Polymeric NPs Characterization

NPs (5 μL) were diluted in 995 μL of H_2_O MilliQ filtered 0.2 μm and then analyzed through Dynamic Light Scattering (DLS). Instead, for morphological analysis, NPs (10 μL) were diluted 1:1 *v*/*v* with H_2_O MilliQ filtered 0.2 μm, and a drop of the sample was then deposited on a carbon screen coated with copper; after evaporation of the excess water, transmission electron microscopy (TEM) analysis was performed.

The NPs encapsulation efficiency was indirectly quantified by exploiting the intrinsic fluorescence of the FITC-BSA (maximum excitation/emission 495/521 nm) or doxorubicin (maximum excitation/emission 470/560 nm). The fluorescence signal corresponding to the unencapsulated compound was subtracted from that relating to the starting amount added. Then, an interpolation analysis with the FITC-BSA or doxorubicin standard curve was performed. The fluorescence signal was acquired with the ChemiDoc Imaging System (Bio-Rad, Hercules, CA, USA). The encapsulation efficiency was extrapolated by setting as 100% reference the fluorescent signal given by the starting amount of the compounds.

For NPs binding/internalization studies, 250,000 Nalm-6 cells were centrifuged at 400 rcf for 5 min and resuspended in 500 μL of culture medium (RPMI-1640, Sigma; supplemented with 10% of Fetal Bovine Serum, Sigma; 100 U/mL Penicillin/0.1 mg/mL Streptomycin, Sigma; 1% L-Glutamine, Sigma). Nalm-6 cells were incubated at 37 °C with increasing amounts (1, 2, and 4 μL) of FITC -BSA (Sigma)-NPs under shaking (800 rpm). At the end of incubation, cells were washed twice in Phosphate-Buffered Saline (PBS) and resuspended with 1% Paraformaldehyde (Sigma) diluted in PBS supplemented with 2% of BSA (Sigma), 0.7 mM CaCl_2_, and 0.7 mM MgCl_2_. The binding/internalization on the surface of cells was evaluated by flow cytometric analysis performed by an Attune^®^ NxT Acoustic Focusing flow cytometer (Thermo Fisher Scientific, Waltham, MA, USA), acquiring 10,000 events; data were analyzed with Attune NxT Software (version 2.7, Thermo Fisher Scientific, Waltham, MA, USA).

### 2.3. MTT Viability Assay

Nalm-6 cells (200,000/200 μL of culture medium, RPMI-1640, Sigma; supplemented with 10% of Fetal Bovine Serum, Sigma; 100 U/mL Penicillin/0.1 mg/mL Streptomycin, Sigma; 1% L-Glutamine, Sigma) were incubated for 24 h at 37 °C under shaking (800 rpm) with free drugs (1μM of doxorubicin, Pfizer) or NPs (1μM of encapsulated drug for doxorubicin-loaded NPs and the same amount for FITC-BSA-NPs). Then, cells were resuspended in 200 μL of clear culture medium. Later, 20 μL of MTT (3-(4,5-Dimethylthiazol-2-yl)-2,5-Diphenyltetrazolium Bromide, MTT, Sigma) was added, and samples were incubated for 4 h at 37 °C under shaking (800 rpm). Samples were then centrifuged for 3 min at 20,000 rcf. The supernatant was discarded, and the deposited crystals were solubilized in 200 μL of Dimethylsulfoxide (DMSO, Sigma). The optical density (OD) was measured at 570 nm with ELISA Reader TECAN Infinite M200. The percentage of viable cells was calculated using untreated cells as a reference for 100% viable cells.

### 2.4. Cell Labeling

Nalm-6 cells were labeled with CellTrace^TM^ Calcein Red-Orange-AM (Thermo-Fisher Scientific, maximum excitation/emission 577/590 nm) according to the manufacturer’s instructions. Afterward, for the CD19 antigen expression analysis, cells were incubated with the primary mouse anti-human CD19 antibody (Immunotools, Gladiolenweg, Friesoythe, Germany, final concentration 5 ng/μL). The secondary Alexa 488-conjugated anti-mouse antibody (2 ng/μL, Invitrogen, Carlsbad, CA, USA) was used to reveal bound antibodies. The cell viability and the antigen expression were evaluated by an Attune^®^ NxT Acoustic Focusing flow cytometer (Thermo Fisher Scientific), acquiring 10,000 events; data were analyzed with Attune NxT Software. The same analysis was performed by immunofluorescence; cells were cytocentrifuged on a slide, and nuclei were stained with 4′,6-diamidino-2-phenylindole (DAPI, Sigma). Slides and images were analyzed, respectively, using fluorescence microscope Nikon Eclipse Ti-E live system and Image-J software (version 2.3.0/1.53f, GNU General Public License, Bethesda, MD, USA).

### 2.5. In Vivo Studies

All experimental procedures involving animals were done after Ministerial Approval 04086.N.SGL.

Zebrafish eggs were placed in E3 Medium supplemented with methylene blue 0.5% and incubated at 28 °C, and 24 h after fertilization (hpf), the eggs were manually dechorionated. Embryos were then placed in E3 Medium supplemented with 1-phenyl 2-thiourea (PTU, Sigma, final concentration 0.2 mM) to inhibit the production of melanin.

#### 2.5.1. Biodistribution Studies

NPs biodistribution studies were performed by injecting NPs (4.6 nL/embryo) in the duct of Cuvier of anesthetized embryos (tricaine, Sigma, final concentration 0.02%) using capillary glasses and a Nanoject II Auto-Nanoliter Injector (Drummond Scientific Co., Broomall, PA, USA). The entire process was conducted using a SteREO Microscope Discovery.V8 (Zeiss, Oberkochen, Germany, UE). At 24 h post-injection (hpi), the biodistribution of the NPs was evaluated using the fluorescence microscope Nikon Eclipse T*i*-E live system; then, images were analyzed with Image-J software.

#### 2.5.2. Xenograft Model

Forty-eight hpf embryos were anesthetized using tricaine (Sigma) and placed on agarose plates, and the excess water was removed to facilitate injection. Then, ~500 cells/embryo were injected in a final volume of 4.6 nL (final concentration 0.1 cell/μL) using Nanoject II Auto-Nanoliter Injector (Drummond Scientific). The localized model was set up by injecting cells in the perivitelline space and the diffused ones by injecting cells in the duct of Cuvier. The entire process was conducted using a SteREO Microscope Discovery.V8 (Zeiss). After cell injection, the embryos were kept at 30 °C and evaluated using the fluorescence microscope Nikon Eclipse T*i*-E live system. Images were analyzed with Image-J software.

## 3. Results

### 3.1. NPs Synthesis and Characterization

In order to evaluate the capability of PLGA-PVA polymeric NPs to be internalized by cells of the leukemia cell line model Nalm-6, the first type of NPs was produced. FITC-BSA-NPs (BSA-NPs) consisted of two parts: the external shell and the aqueous core. The shell represents the outer layer material, and it was composed of PLGA (30 mg/mL) and PVA (2% *w*/*v*). Instead, the core represents the inner material, consisting of FITC-BSA (20 mg/mL in a water buffer). Firstly, BSA-NPs were morphologically characterized, showing round shapes, a diameter lower than 350 nm, and a negative surface charge. The average diameter, polydispersity index (PDI), and zeta potential values of these NPs are reported in Figure 1A.

In vitro tests were performed to assess NPs binding and internalization on malignant B-cells. Different amounts of BSA-NPs were incubated for 1 h with Nalm-6 cells and analyzed by flow cytometry. The binding/internalization of fluorescent BSA-NPs was evaluated by comparing the percentage of positive cells in Figure 1C, demonstrating a dose-related binding/internalization, which increased from 23.44% for 1 μL to 46.29% (2 μL) and to 57.46% for 4 μL. The Mean Fluorescence Intensity (MFI, Figure 1D) values of cells incubated with NPs were compared with the autofluorescence of cells. At 1 h of incubation, low amounts of NPs (1 and 2 μL) interact with cells five times less than the higher amount tested (438 and 459 vs. 2423).

### 3.2. NPs Biodistribution in Healthy ZF

To evaluate the biodistribution of BSA-NPs in ZF, embryos were manually dechorionated 24 h post-fertilization (hpf), and their pigmentation was inhibited using PTU, an enzyme that can affect the conversion of tyrosine to melanin. BSA-NPs were micro-injected into the duct of Cuvier in at least 20 animals per group 48 hpf and were analyzed by fluorescence microscopy over the next three days. BSA-NPs were visualized by following the FITC-BSA encapsulated inside their core (Figure 2).

Twenty-four hours post-injection (hpi), NPs were already visible throughout the ZF blood circulation in the entire bloodstream, and an accumulation of NPs was appreciable in the tail of the fish, where the lower velocity of blood circulation and its flatness facilitates visualization of circulating NPs. Moreover, this area is known as the posterior blood island (PBI), and it is known as a macrophage-rich area [20].

### 3.3. Elimination of NPs

To evaluate the interaction between NPs and ZF’s macrophages, the transgenic line of ZF Tg(mpeg1:mCherry) was employed. This transgenic line possesses red-fluorescent macrophages already visible in the PBI 24 hpf. Firstly, ZF were randomly assigned to three different groups and then analyzed through fluorescence microscopy. A Region Of Interest (ROI, Figure 3A) was drawn in the tail of the fish in correspondence with the PBI; the red fluorescence in this area was quantified in each larvae, and the corrected total cell fluorescence (CTCF) was calculated as follows:CTCF=Integrated Density−[(Area of Selected Cells)×(Mean Fluorescence of Background Readings)]

The data highlight that the three groups of ZF possess comparable amounts of red fluorescence and, as a consequence, comparable amounts of macrophages (Figure 3B). Therefore, 48 hpf, BSA-NPs were injected in the duct of Cuvier of the Tg(mpeg1:mCherry) transgenic line of ZF. As previously described, a ROI was drawn in the tail of the ZF, and fluorescent signals were analyzed. As shown in Figure 3C, in the selected ROI set in the PBI (panel b), the concomitant presence of macrophages (red fluorescence, panel d) and BSA-NPs (green fluorescence, panel e) was detected. The colocalization value was then compared to the fluorescent signal given by free BSA-NPs, highlighting that at 24 hpi, 20% of NPs were eliminated through macrophages, whereas 80% of them were free and still able to exploit their function (Figure 3D).

### 3.4. Setup of a Tumor-Bearing Model of ZF

A localized xenograft model using Nalm-6 cells was initially set up to test whether the tumor burden represented by the number of Nalm-6 cells could be easily determined from the fluorescent signal given by the fluorescent-labeled cells. In particular, B-cells were labeled with the Calcein-AM (Figure 4A), a cell-permeant dye used to determine cell viability in most eukaryotic cells, and their fluorescence was verified through flow cytometry (Figure 4B) and fluorescent microscopy (Figure 4C). Following the experimental timeline (Figure 4D), 48 hpf, ~500 Calcein-AM-labeled Nalm-6 cells were injected into the perivitelline space, resulting in the localization of tumor cells in the ventral thoracic area (Figure 4E).

ZF were followed over time, and the red-fluorescent Calcein-AM emission in live cells was analyzed through fluorescence microscopy (Figure 4E) immediately after cells’ injection (left panels), 24 hpi (central panels), and 48 hpi (right panels). The Nalm-6 tumor mass did not lose fluorescence over time, indicating that the cells were still alive. In addition, the Nalm-6 tumor mass appeared selectively localized in the flank of the ZF, over the yolk sac, and Nalm-6 cells did not appear to move through the ZF body.

### 3.5. Doxorubicin-Loaded NPs Synthesis and Characterization

PLGA-PVA NPs filled with doxorubicin (doxorubicin-NPs) were produced and consisted of two parts, as previously described for BSA-NPs: the external shell and the inner core, which was composed of doxorubicin (25 mg/mL in a water buffer). Doxorubicin-NPs were tested in vitro, showing round shapes, a diameter lower than 300 nm, and a small negative surface charge. The average diameter, polydispersity index (PDI), and zeta potential values of these types of NPs are reported in Figure 5A.

To investigate the capability of drug-loaded NPs to kill Nalm-6 cells, they were tested in a viability assay using MTT evaluation (Figure 5B). Nalm-6 cells were incubated with doxorubicin-NPs (1 μM doxorubicin final concentration) and compared to cells incubated with the free drug (doxorubicin, 1 μM final concentration) or BSA-NPs. As evidenced in Figure 5B, the efficacy of doxorubicin-NPs (1 μM final concentration of the loaded drug) was comparable to that obtained with the free drug, and in parallel, it was demonstrated that this effect was obtained by the encapsulated drug and not due to polymers of the NPs’ structure, because BSA-NPs did not affect cell viability.

### 3.6. Doxorubicin-Loaded NPs Arrest Tumor Growth

To evaluate the therapeutic efficacy of drug-loaded NPs and polymer toxicity, a ZF model of B-cell pathology was developed, with the specific aim to obtain a distributed model and not a localized one. As described in Figure 6A, ~500 Calcein-AM-labeled Nalm-6 cells were injected into the perivitelline space of ZF embryos at 48 hpf. BSA-NPs or doxorubicin-NPs (4,6 nL) were injected into the duct of Cuvier 4 h after the cells took root. ZF were analyzed immediately after the injection of NPs (Figure 6B, upper panels) and 24 hpi (Figure 6B, lower panels) through fluorescence microscopy.

As described in Figure 6C, the CTCF of red-fluorescent Nalm-6 cells was calculated at 0 hpi and compared to the CTCF at 24 hpi. Treatment with BSA-NPs was not able to reduce the signal corresponding to the Nalm-6 tumor burden. In this case, the variation of the fluorescent signal given by NPs highlighted the safety of the polymers in vivo. On the other hand, treatment with doxorubicin-NPs was able to decrease the fluorescent signal corresponding to the Nalm-6 tumor burden (*p* ≤ 0.0001, comparison between BSA-NPs and doxorubicin-NPs). This result demonstrated the ability of doxorubicin-NPs to arrest tumor growth in a localized model of B-cell malignancy.

## 4. Discussion

In this study, a fluorescent-based quantification method was developed to evaluate the number of tumor cell line cells in ZF models. In particular, a localized xenograft model of B-ALL in ZF was developed using Nalm-6 cells. In this context, a variety of features made ZF an excellent model organism. These include primarily economic advantages due to its small size requiring low space and maintenance costs [21]. Moreover, the genome is completely sequenced, shows a high level of similarity with humans (approximately 70%), and is easily manipulated [22]. This made it possible to create transgenic or mutant ZF lines to facilitate the observation of internal structures or biological processes. An example is the Casper mutants, which maintain the body transparency of the embryonic stage until adulthood. There are also reporter lines, such as tg (fli1a:eGFP) and mpeg1.1:mCherry, which specifically respectively label endothelial cells and macrophages [23]. From the reproductive perspective, ZF represents an excellent experimental system since it can lay about 200 eggs per mating. Thus, having a large number of embryos guarantees the possibility of conducting large studies. There are also significant development considerations that exhibit optimal experimental properties [21].

Xenotransplantation represents a way to model tumor development and then study possible therapeutic approaches. ZF embryos represent a powerful model for cancer research, with a growing appreciation for their efficiency. Particularly, the transparent body wall and the absence of the immune response make ZF embryos optimal for xenograft as a tool to evaluate cancer progression and drug screening. Initial evidence of this was reported in 2005, when Lee et al. engrafted a human melanoma cell line in ZF, demonstrating the survival and migration of exogenous cells [23]. Over the years, ZF then acquired importance, joining the common murine models. Although the latter remain the “gold standard”, limitations such as high costs, greater complexity, and the need for immunosuppression for xenograft make this system less flexible [24]. In contrast, there are advantages to the experimental practicality of ZF. Firstly, being transparent, the clear and simple observation of transplanted cells is guaranteed by employing microscopy techniques. Indeed, immediately after the injection, it is possible to observe the embryos under a microscope and follow them over time to evaluate any changes (e.g., tumor mass formation). Moreover, the use of transgenic lines and labeled transplanted cells help the study of the developmental process of the tumor [21]. All these observations demonstrate the interesting role of ZF xenotransplantation in studying tumor development and validating therapy efficacy. Starting from these considerations, ZF embryos can be used to test “next-generation” approaches such as NPs, which owe their success in drug delivery to the possibility of overcoming problems in cancer therapy-related, off-target side effects.

The treatment of pediatric cancers has been a success story, with current OS of ~80% in the United States. Nonetheless, this success has occurred at a significant price; with increased long-term cancer survivorship, there are also side effects. The most relevant is cardiovascular toxicity, which became apparent soon after the widespread use of anthracyclines in the 1970s. Several years after their discovery, these drugs continue to evoke considerable interest in basic science and clinical trial research [25,26,27]. In fact, anthracycline chemotherapy regimens play a prominent role in many cancer treatments, e.g., 50 to 60% of childhood cancer survivors are treated with an anthracycline regimen to the point that anthracyclines are listed among the World Health Organization (WHO) model list of essential medicines [28].

Despite their widely acknowledged efficacy, significant restrictions are associated with anthracycline treatment; the chemotherapy intensity has been raised to the limit of tolerance; therefore, there is a need for novel therapeutic approaches that are able to further improvement in outcomes and reduction in adverse effects [29].

NPs represent an alternative approach that is supposed to be more specific thanks to the possibility of treating the pathology through encapsulated chemotherapy that reaches the desired site, where they release the content.

The nanostructures developed in this study were made of PLGA-PVA polymers, which have been widely investigated to formulate biodegradable devices for the sustained delivery of drugs, proteins, and nucleic acids. Their biodegradability, biocompatibility, and safety profile are some of the main features that make these polymers optimal, even in vivo. PLGA is a copolymer consisting of two different monomer units, poly (glycolic acid) (PGA) and poly (lactic acid) (PLA), linked together, and the result is a linear, amorphous aliphatic polyester product. In vivo, the polymer undergoes degradation by hydrolysis with the following formation of the original monomers (i.e., lactic acid and glycolic acid), which are endogenous monomers also produced in normal physiological conditions. Thus, they are easily processed via metabolic pathways such as the Krebs cycle and removed as carbon dioxide and water, causing minimal systemic toxicity. Its success is particularly related to its continued drug release compared to conventional devices and to the negative charge, which is also crucial because it strongly influences the interaction between NPs and cells [30]. Another synthetic and biocompatible polymer extensively studied is PVA. PVA is frequently used as an emulsifier in the formulation of PLGA-PVA NPs due to its ability to form an interconnected structure with the PLGA, helping to achieve NPs that are relatively uniform and small [31,32]. In addition to the above-mentioned reasons, the simple and reproducible synthesis process and the possibility of surface functionalization (i.e., with targeting mechanisms) led us to focus on these nanodevices. Moreover, PLGA-PVA NPs keep water-soluble drugs/compounds trapped in the aqueous inner core, making these NPs an optimal delivery system.

PLGA-PVA NPs are spherical nano-sized core/shell structures that bind B-cells in a dose-dependent manner. When injected into the bloodstream of ZF embryos through the duct of Cuvier, which is an embryonic vein structure collecting all venous blood and leads directly to the heart’s sinus venosus, NPs are broadly distributed in the embryo’s body with a predilection for a region in the tail of the fish, known as the posterior blood island (PBI). This flat area is characterized by a reduced speed of the bloodstream, which facilitates the visualization of circulating NPs. Since the PBI is known to be a macrophage-rich area [20], the different accumulation of NPs is probably due to the macrophages’ engulfment. To confirm this point, the transgenic ZF line Tg (mpeg1:mCherry) was previously exploited to visualize the interactions between macrophages and other immune cells or pathogens in vivo and to reexamine macrophage roles in inflammation, wound healing, and development, as well as their interactions with other cell types (e.g., vasculature, muscle) in vivo [33]; more interestingly, they represent a useful model in the study of the interaction between macrophages and NPs injected into the bloodstream. This approach allowed us to clarify that ~20% of PLGA-PVA NPs colocalize with the macrophages, indicating that this population of cells is implicated in their elimination, and this is probably due to the high surface-area-to-volume ratio. Moreover, when administered in vivo, NPs interact with circulating proteins (e.g., albumin, complement proteins, fibrinogen, and immunoglobulins) that are attracted to their surface to form a coating layer called protein corona; this affects the biological identity of NPs and, consequently, their functionality [34]. As a consequence, NPs became more visible to phagocytic cells, which recognize the materials via ligand–receptor interactions leading to the rapid elimination of NPs. On the other hand, the remaining ~80% of NPs remain free to execute their duty at the moment of the analysis.

In vivo experiments are strongly recommended to verify NPs’ behavior in a complex environment; the optimal way to assess the efficacy of drug-loaded NPs is represented by observation of tumor growth arrest and a reduction in its burden. Therefore, the ZF embryo represents an interesting animal model considering that it embodies a versatile platform for xenotransplantation without risk of rejection. On the other hand, this feature also represents a limitation. Indeed, due to the lack of an adaptative immune system in this developmental stage, which is a difference from other, more complex animal models, such as mice and rats, ZF embryos do not allow us to evaluate the key role of adaptive immunity in the tumor microenvironment and cancer progression [7].

A diffused model of ALL is obviously a more realistic scenario; however, to better quantify the effect of drug-loaded NPs, a localized one is recommended. Therefore, a localized xenograft model was set up by injecting cells in the perivitelline space (at the margin between the yolk sac and the embryonic cell mass). The model clearly allowed the visualization of tumor cells localized in the ventral thoracic area and the verification of the tumor’s dimensions through the fluorescent signal given by viable cells.

For what concerns the treatment of ALL, anthracyclines are a well-known class of chemotherapeutics that act mainly by intercalating DNA and interfering with DNA metabolism and RNA production. Two major dose-limiting toxicities of anthracyclines include myelosuppression and cardiotoxicity [26]. The PEGylated liposomal doxorubicin formulation “Doxil” was the first US Food and Drug Administration (FDA)-approved liposome chemotherapeutic agent in 1995. It has shown highly selective tumor localization and excellent pharmacokinetic properties in clinical applications. Starting from this point, anthracyclines were chosen as candidates to be encapsulated inside nano-devices. Among anthracyclines, doxorubicin is widely used; however, its clinical use is restricted due to the severe risk to develop cardiotoxicity [26]. To specifically address this point, PLGA-PVA NPs loaded with doxorubicin were produced; their in vitro characterization demonstrated that the features of the NPs were not influenced by the presence of the drug in the core and that the cytotoxic activity of the drug was also maintained after its loading into the nano-devices.

The therapeutic efficacy of drug-loaded NPs was easily verified in a localized model of human leukemia in ZF embryos. It was possible to immediately visualize the local distribution of the cells and to measure the tumor burden. The potential therapeutic effect of drug-loaded NPs was obtained in only 24 h. Tumor-bearing ZF embryos were analyzed immediately after the injection of NPs and 24 hpi through fluorescence microscopy. The variation of the fluorescent signal highlights that the tumor burden in ZF that received drug-loaded NPs was reduced by more than 85% in comparison with ZF treated with NPs. Therefore, these data confirm the safety of the polymers in vivo and the therapeutic efficacy of drug-loaded NPs in a localized model of B-cell malignancy.

## 5. Conclusions

In this paper, we investigated ZF as a 3.0 animal model, resulting in a useful tool to study and characterize polymeric NPs. Our results highlighted that ZF can have a pivotal role in the study of the biodistribution, elimination, and therapeutic efficacy of doxorubicin-loaded polymeric NPs, which represent a promising tool for the treatment of B-cell malignancies.

More generally, ZF embryos can be suitable for the study of other nano-carriers to evaluate their toxicity, analyze their specificity, easily compare different therapeutic approaches, and choose the most effective. ZF represents a relevant model of B-cell malignancies, but the same approach can be transferred to the development of human–ZF models of other tumors, which could be useful in the development of specific drug-loaded polymeric NPs.

## Figures and Tables

**Figure 1 biomedicines-10-02252-f001:**
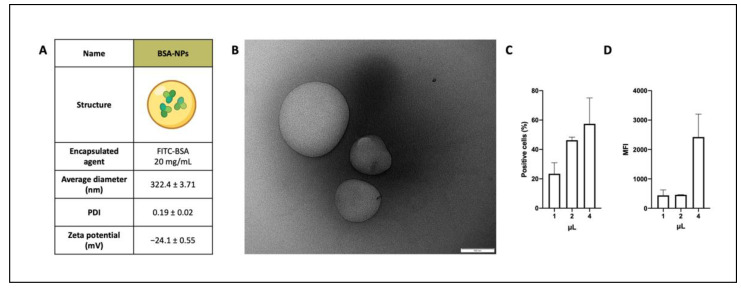
NPs’ characterization: (**A**) Physicochemical characteristics of BSA-NPs, which are composed of PLGA-PVA polymers and filled with FITC-conjugated BSA. (**B**) TEM image of NPs (scale bar 100 nm). Nalm-6 cells were incubated with different amounts of NPs and analyzed by flow cytometry to evaluate their binding/internalization. (**C**) The percentage of positive cells and (**D**) Mean Fluorescence Intensity (MFI) are reported in the histograms. Data are reported as the mean ± SD.

**Figure 2 biomedicines-10-02252-f002:**
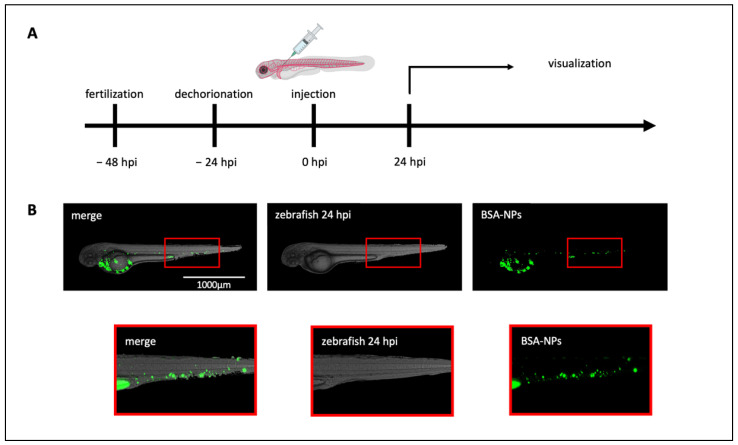
Zebrafish as an animal model for biodistribution studies: (**A**) Timeline of the experimental procedure. ZF were followed for a total of 5 days. The reference point was set at the moment of the injection (0 h post-injection, hpi), which was performed 48 h post-fertilization. Embryos were manually dechorionated and incubated with PTU to prevent the formation of pigmented areas in the body 24 h prior to the injection (−24 hpi). (**B**) BSA-NPs were injected in the duct of Cuvier and analyzed 24 hpi demonstrating an accumulation of fluorescent BSA-NPs in the tail of ZF, known as posterior blood island (PBI), which is a macrophage-rich area. Magnification 40×. Scale bar 1000 μm.

**Figure 3 biomedicines-10-02252-f003:**
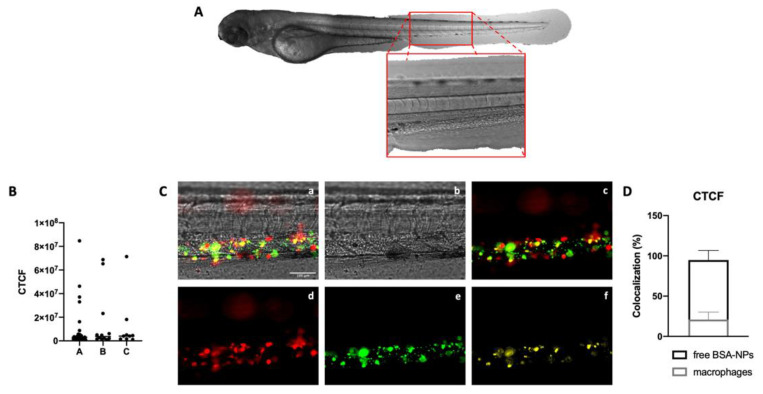
Zebrafish as a model for elimination studies: (**A**) Representation of the ROI chosen for the analysis of fluorescent signals in the Tg(mpeg1:mCherry) given by the presence of BSA-NPs. (**B**) Data analysis of the CTCF of the red-fluorescent areas (macrophages) of zebrafish randomly assigned to three different groups, suggesting that the fluorescent signal of macrophages is comparable between different animals. Data are reported as the median. (**C**) BSA-NPs were injected in the duct of Cuvier of transgenic line Tg(mpeg1:mCherry), and 24 hpi, ZF were analyzed through fluorescence microscopy. The upper left panel (**a**) represents the merge between macrophages (red, **d**), BSA-NPs (green, **e**), and the zebrafish’s tail (gray, **b**). In the merge (**c**) between macrophages (red, **d**) and BSA-NPs (green, **e**), it is possible to observe yellow spots given by their colocalization, better appreciable isolating them (**f**), representing BSA-NPs engulfed by macrophages. (**D**) Data analysis of the CTCF of the colocalized areas. All data are reported as the mean ± SD. Magnification 200×. Scale bar 100 μm.

**Figure 4 biomedicines-10-02252-f004:**
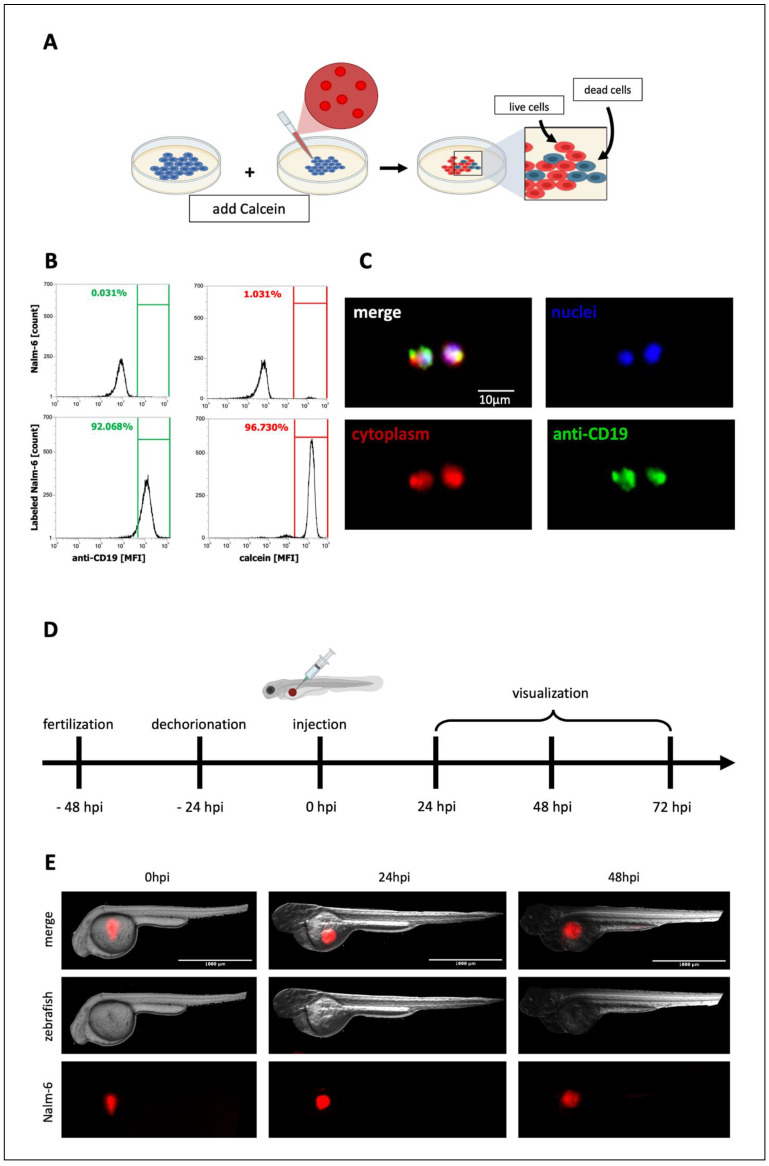
Localized xenograft zebrafish model: (**A**) Nalm-6 cells were labeled with Calcein-AM. After the internalization of Calcein-AM, the non-fluorescent Calcein-AM is converted in living cells to red-fluorescent Calcein-AM. Nalm-6 cells were also labeled with DAPI and anti-CD19 antibody (as described in materials and methods) and analyzed by (**B**) flow cytometric analysis and (**C**) immunofluorescence microscopy. Magnification 200×. Scale bar 10 μm. (**D**) Timeline of the experimental procedure. (**E**) Calcein-AM-labeled Nalm-6 cells were injected at 48 hpf in the perivitelline space of larvae. ZF were analyzed immediately after the injection (left panels), 24 hpi (central panels), and 48 hpi (right panels). Magnification 40×. Scale bar 1000 μm.

**Figure 5 biomedicines-10-02252-f005:**
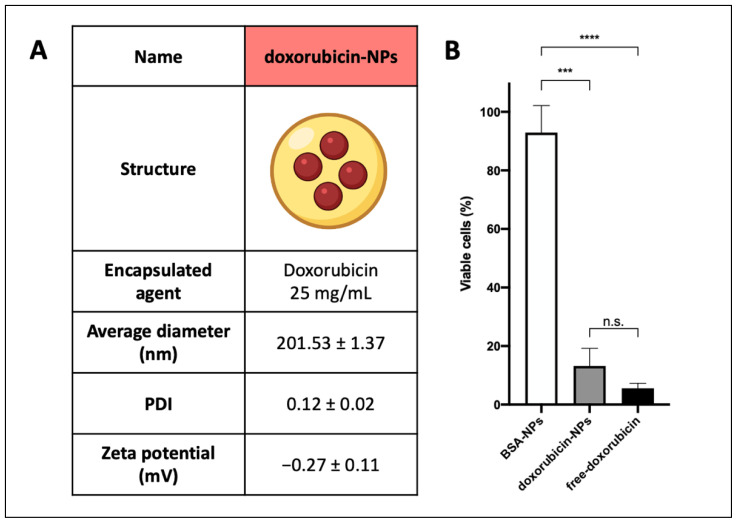
Drug-loaded NPs: (**A**) Physicochemical characteristics of doxorubicin-NPs, which are composed of PLGA-PVA polymers and filled with doxorubicin. (**B**) Nalm-6 cells were incubated with free doxorubicin, doxorubicin-NPs, or BSA-NPs for 24 h. Samples were then analyzed by evaluating MTT assay. Data are reported as the mean ± SD. *p* ≤ 0.001 = ***, *p* ≤ 0.0001 = ****, n.s. not significant.

**Figure 6 biomedicines-10-02252-f006:**
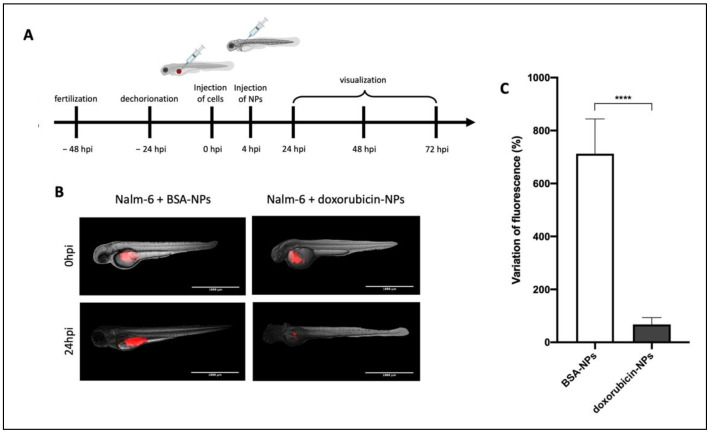
Evaluation of the therapeutic effect of doxorubicin-NPs: (**A**) Timeline of the experimental procedure. (**B**) Nalm-6 cells (labeled with red-fluorescent Calcein-AM) were injected into the perivitelline space of zebrafish embryos 48 hpf, and 4 h later, BSA-NPs (left panels) or doxorubicin-NPs (right panels) were injected in the duct of Cuvier. Zebrafish were analyzed immediately after the injection of NPs (upper panels) and 24 hpi (lower panels). Magnification 40×. Scale bar 1000 μm. (**C**) Data analysis of the variation of cells’ fluorescence after treatment with NPs expressed as a percentage of untreated animals. All data are reported as the mean ± SD. *p* ≤ 0.0001 = ****.

## Data Availability

Data are contained within the article.

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
