# Peer review of "Zebrafish: A Useful Animal Model for the Characterization of Drug-Loaded Polymeric NPs"

_biomedicines, 2022, doi:10.3390/biomedicines10092252_

Round 1

Reviewer 1 Report

The paper by Bozzer et.al., is interesting in this field. I have a few minor suggestions, 

1. How can we avoid the in vivo zebrafish toxicity issue during administration, did the author find anything related to this aspect?

2. What is the circulation time of the delivered nanoparticles?

3. If the authors can verify the cell toxicity findings in vitro would be highly appreciated?

Author Response

Response to Reviewer 1 Comments

We are grateful to the Reviewer for finding our work “interesting in this field” and we thank the Reviewer for his/her comments and suggestions, which allow us to clarify and improve the quality of the manuscript.

Point 1: How can we avoid the in vivo zebrafish toxicity issue during administration, did the author find anything related to this aspect?

Response 1: Thanks for this comment. Zebrafish is an interesting animal model, also for this reason. Indeed, zebrafish embryos are susceptible to developmental disorders and/or changes, such as spontaneous movements and heart morphology, that can be employed as indicators of the acute toxicity of tested compounds. We do not observe these changes during our tests, indicating that the nano-systems are safe.

Point 2: What is the circulation time of the delivered nanoparticles?

Response 2: From the data in our hand, studying these or other nano-systems, the calculated amount of circulating nanostructures decreases about 30% in 24h. However, we want to highlight that, conversely to other animal models (such as mice, rats, etc.), this animal model does not allow a direct quantification of the delivered NPs.

Point 3: If the authors can verify the cell toxicity findings in vitro would be highly appreciated?

Response 3: Thanks for this suggestion. Our toxicological tests were focalized on tumoral cells (Nalm-6 cells) and the results are reported in Figure 5. For what concerns the toxicity of the NPs in other organs and/or tissues, we do not observed modifications in the development of the embryos, either in animals treated with BSA-NPs or doxorubicin-NPs.

Reviewer 2 Report

In the submitted manuscript “Zebrafish: a Useful Animal Model for the Characterization of Drug-Loaded Polymeric-NPs” zebrafish model was used as an in-vivo model to study biodistribution and anti-tumor efficacy of polymeric nanoparticles. The authors have characterized two different polymeric nanoparticles and studied it using leukemia xenograft in zebrafish

The manuscript is well written but requires a major revision before accepting for publications.

1.     Provide how encapsulation efficiency was calculated. Also explain the composition of NP core and shell, where the drug is entrapped?

2.     Also provide some more explanation why these PLGA, PVA were selected for NP synthesis.

3.     In the internalization/ binding experiment, a comparison between BSA-FITC alone and NP-BSA is required.

4.     Provide a rational explanation why the BSA Nanoparticles were used for biodistribution experiment. Also, a comparison is required with BSA-FITC distribution.

5.     Also, doxorubicin has a fluorophore, why the author could not do biodistribution with the doxorubicin NPs itself instead of conducting with BSA-NPs? Please clarify.

6.     In the biodistribution study, there is a significant engulfment of NP by macrophage. Is this because of the size of nanoparticles which are around 300 nm?

7.     Also, the author claims 80 percent of Nanoparticles are free to bind to target even after macrophage engulfment. This might be true for this model but in practical most of the NP gets accumulated in liver, spleen, and kidney in in-vivo models such as mice. How this information from this model can be translated in pre-clinical and clinical settings?

8.     One of the major limitations of this model is it cannot capture the tumor microenvironment since there is a lack of adaptive immunity. Provide some approaches to address these challenges in discussion.

Author Response

Response to Reviewer 2 Comments

We are grateful to the Reviewer for his/her comments and suggestions, which allow us to clarify and improve the quality of the manuscript.

Point 1: Provide how encapsulation efficiency was calculated. Also explain the composition of NP core and shell, where the drug is entrapped? 

Response 1: To address the issue, a new section (2.1) was included to better explain the NPs-synthesis process (page 2, lines 70-86). Moreover, as suggested by the Reviewer, NPs encapsulation efficiency methods have been included in section 2.2 (ex 2.1 - page 2, lines 87-115) of the revised version of the manuscript. Moreover, several sentences discussing the NPs core and shell structure have been included in sections 3.1 (page 4, lines 196-201) and 3.5 (page 8, lines 293-295).

Point 2: Also provide some more explanation why these PLGA, PVA were selected for NP synthesis.

Response 2: As suggested by the Reviewer, several sentences have been now included in the discussion (pages 10-11, lines 389-411) highlighting the main reasons why PLGA and PVA were selected for NPs synthesis.

Point 3: In the internalization/ binding experiment, a comparison between BSA-FITC alone and NP-BSA is required. 

Point 4: Provide a rational explanation why the BSA Nanoparticles were used for biodistribution experiment. Also, a comparison is required with BSA-FITC distribution.

Responses 3 and 4: We thank the Reviewer to have pointed out this aspect of our work.

During the synthesis process, FITC-BSA is encapsulated in the aqueous inner core and retained only inside the NPs, making them easily visible and optimal for biodistribution studies. Moreover, the data in our hands confirms that the FITC-BSA was not released in the solution; however, NPs were also washed to remove the minimum eventually-leaked material.

On these bases, internalization/binding studies, as well as the biodistribution studies, were not performed with BSA-FITC alone.

Point 5: Also, doxorubicin has a fluorophore, why the author could not do biodistribution with the doxorubicin NPs itself instead of conducting with BSA-NPs? Please clarify.

Response 5: As suggested by the Reviewer, both doxorubicin and FITC-BSA have fluorescent features; however, biodistribution studies were performed with FITC-BSA NPs because the goal was to study where NPs are located, without having toxicity related to the drug.

Point 6: In the biodistribution study, there is a significant engulfment of NP by macrophage. Is this because of the size of nanoparticles which are around 300 nm?

Response 6: As suggested by the Reviewer, NPs’ size strongly influences their engulfment, but it is not the only parameter. This section of the discussion has been revised including also a new citation (page 11, lines 427-435).  

Point 7: Also, the author claims 80 percent of Nanoparticles are free to bind to target even after macrophage engulfment. This might be true for this model but in practical most of the NP gets accumulated in liver, spleen, and kidney in in-vivo models such as mice. How this information from this model can be translated in pre-clinical and clinical settings?

Response 7: Of course, as suggested by the Reviewer, when administered in mammal systems, such as mice, NPs gets accumulated in the liver, spleen, and kidney giving interesting information for pre-clinical and clinical settings. However, these cannot be referred to ZF larvae: ZF represents an excellent model for obtaining preliminary information, such as the choice of the optimal nanocarrier as well as drugs or targeting mechanisms. Consequently, the collection of these data in this ZF model allows for reducing the number of mice in the subsequent experimentation stages.

Point 8: One of the major limitations of this model is it cannot capture the tumor microenvironment since there is a lack of adaptive immunity. Provide some approaches to address these challenges in discussion.

Response 8: As suggested by the Reviewer, ZF adaptive immunity is immature until 4-6 weeks post-fertilization, and this feature represents a limitation of this animal model. This section of the discussion has been revised including also a new citation (page 11, lines 436-444). 

Round 2

Reviewer 2 Report

Thanks to the author. The comments are well addressed.